# External validation of a mobile clinical decision support system for diarrhea etiology prediction in children: A multicenter study in Bangladesh and Mali

Stephanie Chow Garbern[1]*[†], Eric J Nelson[2][†], Sabiha Nasrin[3], Adama Mamby Keita[4], Ben J Brintz[5], Monique Gainey[6], Henry Badji[4], Dilruba Nasrin[7], Joel Howard[8], Mami Taniuchi[9], James A Platts-Mills[9], Karen L Kotloff[10], Rashidul Haque[3], Adam C Levine[1], Samba O Sow[4], Nur Haque Alam[3], Daniel T Leung[11]*

[1]Department of Emergency Medicine, Alpert Medical School, Brown University, Providence, United States; [2]Department of Pediatrics, and Environmental and Global Health, Emerging Pathogens Institute, University of Florida, Gainesville, United States; [3]International Centre for Diarrhoeal Disease Research, Dhaka, Bangladesh; [4]Centre for Vaccine Development, Bamako, Mali; [5]Division of Epidemiology, University of Utah, Salt Lake City, United States; [6]Rhode Island Hospital, Providence, United States; [7]Center for Vaccine Development and Global Health, University of Maryland School of Medicine, Baltimore, United States; [8]Department of Pediatrics, University of Kentucky Medical School, Lexington, United States; [9]Division of Infectious Diseases and International Health, University of Virginia, Charlottesville, United States; [10]Department of Pediatrics, University of Maryland, Baltimore, United States; [11]Division of Infectious Diseases, University of Utah School of Medicine, Salt Lake City, United States

*For correspondence:
sgarbern@brown.edu (SCG);
daniel.leung@utah.edu (DTL)

[†]These authors contributed equally to this work

## Abstract

**Background:** Diarrheal illness is a leading cause of antibiotic use for children in low- and middle-income countries. Determination of diarrhea etiology at the point-of-care without reliance on laboratory testing has the potential to reduce inappropriate antibiotic use.

**Methods:** This prospective observational study aimed to develop and externally validate the accuracy of a mobile software application ('App') for the prediction of viral-only etiology of acute diarrhea in children 0–59 months in Bangladesh and Mali. The App used a previously derived and internally validated model consisting of patient-specific ('present patient') clinical variables (age, blood in stool, vomiting, breastfeeding status, and mid-upper arm circumference) as well as location-specific viral diarrhea seasonality curves. The performance of additional models using the 'present patient' data combined with other external data sources including location-specific climate, data, recent patient data, and historical population-based prevalence were also evaluated in secondary analysis. Diarrhea etiology was determined with TaqMan Array Card using episode-specific attributable fraction (AFe) >0.5.

**Results:** Of 302 children with acute diarrhea enrolled, 199 had etiologies above the AFe threshold. Viral-only pathogens were detected in 22% of patients in Mali and 63% in Bangladesh. Rotavirus was the most common pathogen detected (16% Mali; 60% Bangladesh). The present patient+ viral seasonality model had an AUC of 0.754 (0.665–0.843) for the sites combined, with calibration-in-the-large $\alpha$ = −0.393 (−0.455—0.331) and calibration slope $\beta$ = 1.287 (1.207–1.367). By site, the

present patient+ recent patient model performed best in Mali with an AUC of 0.783 (0.705–0.86); the present patient+ viral seasonality model performed best in Bangladesh with AUC 0.710 (0.595–0.825).

**Conclusions:** The App accurately identified children with high likelihood of viral-only diarrhea etiology. Further studies to evaluate the App's potential use in diagnostic and antimicrobial stewardship are underway.

**Funding:** Funding for this study was provided through grants from the Bill and Melinda GatesFoundation (OPP1198876) and the National Institute of Allergy and Infectious Diseases (R01AI135114). Several investigators were also partially supported by a grant from the National Institute of Diabetes and Digestive and Kidney Diseases (R01DK116163). This investigation was also supported by the University of Utah Population Health Research (PHR) Foundation, with funding in part from the National Center for Advancing Translational Sciences of the National Institutes of Health under Award Number UL1TR002538. The content is solely the responsibility of the authors and does not necessarily represent the official views of the National Institutes of Health. The funders had no role in the study design, data collection, data analysis, interpretation of data, or in the writing or decision to submit the manuscript for publication.

## Introduction

Diarrheal diseases remain a leading cause of morbidity and mortality in children younger than five years worldwide, with approximately one billion episodes and 500,000 deaths annually. *James et al., 2018*; *Troeger et al., 2018*. While a significant problem in all countries, the greatest burden of pediatric diarrhea exists in low- and middle-income countries (LMICs), primarily in South Asia and sub-Saharan Africa. *Troeger et al., 2018*. Although the majority of diarrhea episodes are self-limiting and the mainstay of diarrhea treatment is rehydration, clinicians must also make decisions regarding appropriate use of diagnostics and for antibiotic prescribing. Guidelines from the World Health Organization (WHO) recommend against antibiotic use for the treatment of pediatric diarrhea, except for specific presentations of diarrhea such as suspicion of *Vibrio cholerae (V. cholerae)* with severe dehydration, blood in stool, or concurrent illness such as severe malnutrition *World Health Organization, 2005a*. For the majority of diarrhea etiologies, antibiotics are not recommended, particularly for viral causes of diarrhea in which antibiotics have no benefit. *Bruzzese et al., 2018*. Viral pathogens such as rotavirus, sapovirus, and adenovirus, are among the top causes of diarrhea in young children in LMICs, as shown in two large multi-center studies from LMICs, the Global Enteric Multicenter Study (GEMS) and the Malnutrition and Enteric Disease (MAL-ED) study *Kotloff et al., 2013*; *Platts-Mills et al., 2018*.

Laboratory testing by culture or molecular assays are often impractical when treating children with diarrhea in the majority of LMIC clinical settings due to time and resource constraints *Bebell and Muiru, 2014*. As a result, clinicians often make decisions regarding antibiotic use based on non-evidence-based assumptions or broad syndromic guidelines *Kotloff, 2017*. Unfortunately, physician judgment has been shown to poorly predict etiology and need for antibiotics in diarrheal infections. For example, patients presenting to Kenyan hospitals with diarrhea showed that syndrome-based guidelines for *Shigella* infection led to the failure to diagnose shigellosis in nearly 90% of cases *Pavlinac et al., 2016*. Accurate and cost-effective tools to better determine diarrhea etiology at the point-of-care without relying on laboratory tests are greatly needed to reduce antibiotic overuse while conserving scarce healthcare resources.

Electronic clinical decision support systems (CDSS) incorporating prediction models may offer a solution to the challenges of determining diarrhea etiology in low-resource settings. CDSSs have been used in high-income country (HIC) settings to improve the accuracy of diagnosis and reduce costs by avoiding unnecessary diagnostic tests at the point-of-care *Bright et al., 2012*. CDSSs, especially as mHealth applications on smartphone mobile devices, hold great potential for implementing sophisticated clinical prediction models that would otherwise be impossible for providers to calculate manually. These tools can also enable flexibility by clinician choice or automation to optimize the clinical algorithm based on epidemiologic and clinical factors dominant in a given location. Despite opportunities to improve clinical care in a cost-aware mindset, there remains a paucity of data on the

**eLife digest** Diarrhea is one of the most common illnesses among children worldwide. In low- and middle-income countries with limited health care resources, it can be deadly. Diarrhea can be caused by infections with viruses or bacteria. Antibiotics can treat bacterial infections, but they are not effective against viral infections.

It can often be difficult to determine the cause of diarrhea. As a result, many clinicians just prescribe antibiotics. However, since diarrhea in young children is often due to viral infections, prescribing unnecessary antibiotics can cause children to have side effects without any benefit. Excessive use of antibiotics also contributes to the development of bacteria that are resistant to antibiotics, making infections harder to treat.

Scientists are working to develop mobile health tools or 'apps' that may help clinicians identify the cause of diarrhea. Using computer algorithms to analyze information about the patient and seasonal infection patterns, the apps predict whether a bacterial or viral infection is the likely culprit. These tools may be particularly useful in low- or middle-income country settings, where clinicians have limited access to testing for bacteria or viruses.

Garbern, Nelson et al. previously built an app to help distinguish cases of viral diarrhea in children in Mali and Bangladesh. Now, the researchers have put their app to the test in the real-world in a new group of patients to verify it works. In the experiments, nurses in Mali and Bangladesh used the app to predict whether a child with diarrhea had a viral or non-viral infection. The children's stool was then tested for viral or bacterial DNA to confirm whether the prediction was correct. The experiments showed the app accurately identified viral cases of diarrhea.

The experiments also showed that customizing the app to local conditions may further improve its accuracy. For example, a version of the app that factored in seasonal virus transmission performed the best in Bangladesh, while a version that factored in data from recent patients in the past few weeks performed the best in Mali. Garbern and Nelson et al. are now testing whether their app could help reduce unnecessary use of antibiotics in children with diarrhea. If it does, it may help minimize antibiotic resistance and ensure more children get appropriate diarrhea care.

use of CDSS for infectious disease etiology determination and to support appropriate antibiotic use in LMICs *Tuon et al., 2017*.

Our team previously derived and internally validated a series of clinical prediction models using data from GEMS, integrating characteristics of the present patient's diarrhea episode (patient-specific factors including age, blood in stool, vomiting, breastfeeding status, and mid-upper arm circumference) with external data sources (such as characteristics of recent patients, historical prevalence, climate, and seasonal patterns of viral diarrhea 'viral seasonality') in a modular approach *Brintz et al., 2021*. The best-performing model, which used 'present patient'+ location-specific viral seasonality data, had an internally validated area under the receiver-operating characteristic curve (AUC) of 0.83 *Brintz et al., 2021*. The objective of this study was to prospectively externally validate the models for the prediction of viral-only etiology of diarrhea in children aged 0–59 months in Bangladesh and Mali and demonstrate a proof-of concept for the incorporation of the primary model ('present patient'+ location-specific viral seasonality) into a mobile CDSS software application ('App') for use in LMIC settings with high diarrheal disease burden.

## Materials and methods
### Study design and setting
A prospective, observational cohort study was conducted in Dhaka, Bangladesh and Bamako, Mali. Enrollment was conducted in Bangladesh at the Dhaka Hospital of the International Center for Diarrhoeal Disease Research, Bangladesh (icddr,b) rehydration (short stay) unit and in Mali at the Centres de Santé de Référence (CSREF) and the Centres de Santé Communautaires (CSCOM) in Commune V and VI in Bamako, Mali. These locations were selected because of their geographic proximity to GEMS study sites from which the clinical prediction models were trained, without using the same sites. Participants were enrolled in Bangladesh during November and December 2019 and Mali during

January and February 2020. The Dhaka Hospital of the icddr,b provides free clinical services to the population of the capital city of Dhaka, Bangladesh and surrounding rural districts and cares for over 100,000 patients with acute diarrhea each year. The CSREF and CSCOM of communes V and VI in Mali serve a catchment area of 2 million people. CSCOM provides basic care such as family planning, vaccination and outpatients visits, while patients with severe illness are referred to the CSREF where there is capacity for hospital admission for medical conditions and for basic and intermediate surgeries.

## Study participants and inclusion/exclusion criteria

Patients under five years of age (0–59 months) with symptoms of acute diarrhea were eligible for enrollment. Acute diarrhea was defined as three or more loose stools per day for less than seven days. Patients were excluded using the following criteria: no parent or primary caretaker available for consent, diarrhea lasting seven days or longer, fewer than three loose stools in the prior 24 hr, or having a diagnosis of severe pneumonia, severe sepsis, meningitis, or other condition aside from gastroenteritis.

## Staff training and oversight

General practice nurses were hired specifically to collect data at both study sites, and study nurses had no other patient care responsibilities during the study period. Nurses received training in study procedures under the guidance of the research investigators. Training topics included: screening procedures, obtaining informed consent, collecting clinical data and laboratory procedures. Study nurses also received practical hands-on training regarding the use of the App to ensure all nurses were comfortable with entering data and using the devices during clinical workflow.

## Data sources and processing, model development and internal validation

'Modular' clinical prediction models for the outcome of viral etiology of pediatric diarrhea were previously derived and internally validated with full details previously published by Brintz et al in 2021 *Brintz et al., 2021*; *Brintz et al., 2020*. Briefly, a series of five models predicting viral etiology were independently developed based on the hypothesis that including location-specific 'external' data sources (i.e. characteristics such as recent patients or climate data in addition to the present patient's characteristics), may improve predictive performance. This study team's prior work described the development of these predictive models that integrates multiple sources of data in a principled statistical framework using a 'post-test odds formulation'. This method incorporates observed prior knowledge of a prediction, typically using prior known location-specific prevalence (e.g. historical prevalence of viral diarrhea in Mali), as pre-test odds and then updates the pre-test odds using a single model or series of models based on current data. It also enables electronic real-time updating and flexibility in a 'modular' fashion, such that the component models can be flexibly included or excluded according to data availability, an important consideration for LMIC settings in which prior epidemiologic data

**Table 1.** Model terminology definitions and descriptions.

| Model name | Description and features included |
| --- | --- |
| Present patient | Random forest variable importance screening was used to screen variables for fitting a logistic regression model from the GEMS data including only five clinical variables (selected from candidate variables which would be accessible to clinicians at the point-of-care) *Brintz et al., 2021*. The five variables include: age, blood in stool (yes/no), vomiting (yes/no), breastfeeding status (yes/no), and mid-upper arm circumference (MUAC; as measured in cm) |
| Viral seasonality | This model included the standardized seasonal sine and cosine curves modeling the country-specific seasonal patterns of viral diarrhea |
| Climate | This model included rain and temperature averages using a two-week aggregation of the five nearest National Oceanic and Atmospheric Administration (NOAA)-affiliated weather stations to the hospital sites. |
| Historical patient (Pre-test odds) | Pre-test odds were generated using historical rates of viral diarrhea by site and date using data from the GEMS study. |
| Recent patient (Pre-test odds) | Pre-test odds were generated using data from patients in the prior four weeks. |

may be unavailable. The post-test odds formulation combines the likelihood ratios derived from these independent models along with pre-test odds into a single prediction. In order to externally validate the predictions from the post-test odds, we processed the data from this study to match the variables used in previously trained models as closely as possible. *Table 1* shows the terminology used to refer to each model and the features included in each model.

The derived models used clinical, historical, anthropometric and microbiologic data from the GEMS study, a large case-control study conducted at seven sites in Asia and Africa (The Gambia, Kenya, Mali, Mozambique, Bangladesh, India, and Pakistan) which enrolled 22,568 children under 5 years, including 9439 children with moderate/severe diarrhea and 13,129 controls *Kotloff et al., 2013*. Demographics, predictors and viral-only outcome data from the development datasets from GEMS in Bangladesh and Mali are shown in *Supplementary file 1*. Additional location-specific sources of data used for model development included local climate (i.e. weather) data, and site-specific viral diarrhea seasonality modeled using sine and cosine curves ('viral seasonality'). Pre-test odds were generated using epidemiologic data based on historical prevalence from the same study site ('historical patient') and from the past 4 weeks ('recent patient') at the same study site.

More specifically, local weather data proximate to each site's health centers was obtained using the National Oceanic and Atmospheric Administration (NOAA) Integrated Surface Database. Climate model features include rain and temperature averages based on a 2-week aggregation of the inverse-distance weighted average of the nearest five NOAA-affiliated weather stations to the hospital sites. Weather stations at a distance of greater than 200 km were excluded. Standardized seasonal sine and cosine curves with a periodicity of 1 year, $sin\left(\frac{2\pi t}{365.25}\right)$ and $cos\left(\frac{2\pi t}{365.25}\right)$, where $t$ is based on the date, were used to model the location-specific seasonal patterns of viral etiology of diarrhea *Brintz et al.,*

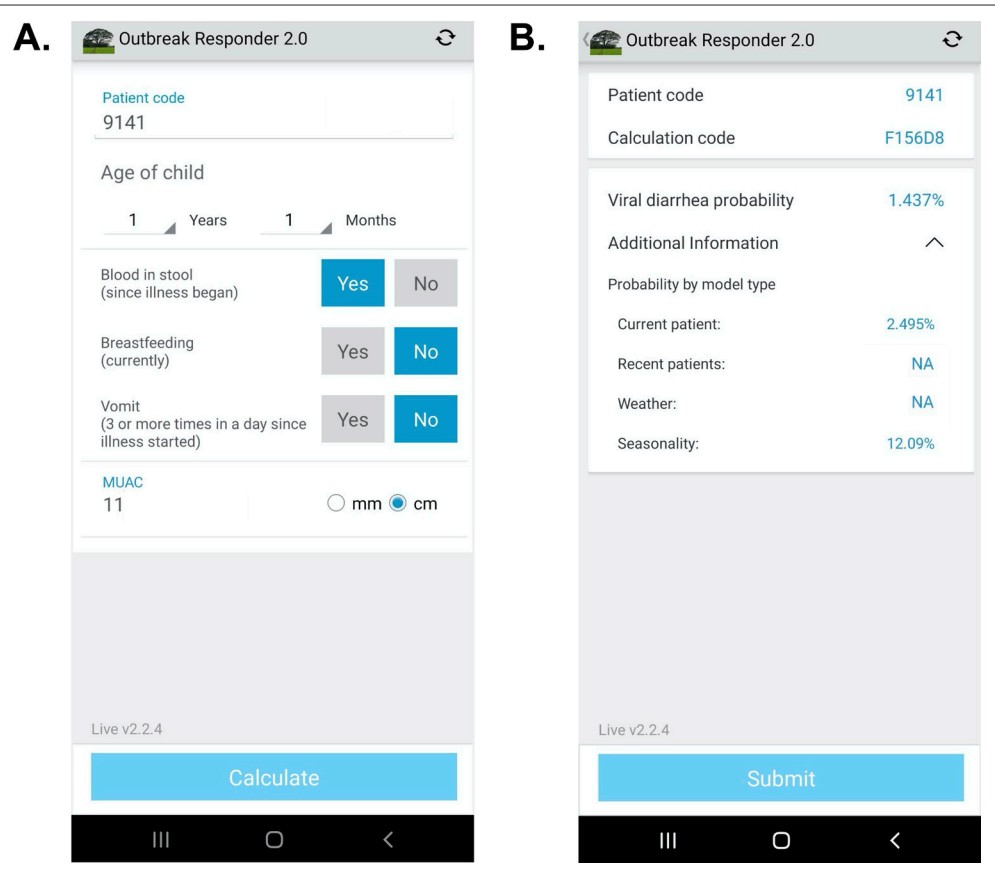

**Figure 1.** App user interface. (**A**) Input page after application launch.( **B**) Output page with an example showing calculated probability of viral-only diarrhea. The '^' symbol represents an open accordion menu with the component probabilities. 'Current patient' refers to the present patient model. 'Weather' (climate) and 'recent patients' (pre-test odds) were not active in this configuration.

*2021*. The seasonal sine and cosine values as well as temperature and rain averages (climate) were calculated for the dates in this study as described previously *Brintz et al., 2021*.

## Technology development

Software architecture platform and user interface concepts for the App were derived from two prior related studies *Khan et al., 2020*; *Haque et al., 2017*. The design herein was intended to demonstrate a 'live' proof-of-concept that the models could be successfully configured as a CDSS on a mobile device for high-volume clinical settings in an LMIC. The App was configured in English (Bangladesh) and French (Mali). On the input page, clinical variables were restricted to those required for the App (*Figure 1*). Text was favored over symbology to increase clarity based on prior experience. Computations were initiated locally on the device after pressing 'calculate'. On the output page, the patient code and randomized calculation code were used to enable joining the digital record to the paper case report form. The design objective was to identify weaknesses in the UI for data entry using paper data entry as the reference standard, and address weaknesses after the study. The codebase was allowed to be iterated once at the transition between the Bangladesh and Mali phases of the study to address engineering challenges exposed during 'live' deployment. The probability of a viral-alone etiology was provided, and the probabilities by model type were accessible via a dropdown menu. Data were encrypted for security on the device and upon transfer to a HIPAA compliant server. The code bases consisted of Python (server), Java (web portal interface), Android operating system, and TensorFlor Lite (TFLite) on the Android device. TFLite is an open-source cross-platform deep-learning framework. TFLite converts pre-trained models from Python in TensorFlow to a format that is optimized for speed to run models and store data on mobile devices. Therefore, all calculations were performed on-device (locally on the smartphone) and allowed App use irrespective of internet connectivity. Calculated probabilities were displayed near instantaneously (within 1 s). The App deployed a single primary model (present patient+ viral seasonality as described above) which used data collected at the point-of-care and input by the clinician user, and viral seasonality curves; climate and recent patient data sources were not included because they did not add to the AUC at the GEMS study sites. The models were not re-trained on the server after deployment for this validation study in order to align with standards of clinical practice for AI-enabled clinical decision support currently being set by Boards of Medical Informatics; according to these standards, validated models derived using machine learning should not be subsequently iterated without validation *AMIA, 2021* For future iterations of the software that use models that require near real-time data input (e.g. climate data, recent patient data), the local database on the device was programmed to automatically fetch climate data and recent patient data (no PHI) from the server if the server was available. The device would also check the server for updated data on a scheduled basis (hourly) and if needed, manually fetch data when desired by the user.

## Study procedures

In Bangladesh, due to the high volume of potentially eligible patients presenting daily to icddr,b Dhaka Hospital, study staff randomly selected participants for enrollment on arrival 9am-5pm Sunday to Thursday. Random selection was accomplished using a black pouch filled with white and blue marbles in a preset ratio. Study nurses drew a marble each time a patient presented to the rehydration unit. If a blue marble was pulled, the patient was screened for inclusion and exclusion criteria as described above. After each marble was pulled, it was returned to the bag and shaken. An average of approximately eight patients were enrolled per working day which allowed for high-quality data collection and integrity of all study protocols to be maintained. In Mali, a consecutive sample of patients presenting with acute diarrhea were enrolled. After initial assessment by the facility doctor, children with acute diarrhea were referred to the study team for screening. Study staff were located in the intake area and potential participants' information was recorded in a screening log. All patients presenting with diarrhea were assessed for eligibility. After screening, research staff provided the parent or guardian with information about the purpose of the study, risks and benefits in Bangla (Bangladesh) and Bambara (Mali) language. Research staff then obtained written consent if the parent or guardian agreed to participate on behalf of the child. In cases where the parent or guardian was illiterate, the consent form was marked with a thumbprint for signature, based on standards for informed consent at icddr,b and CVD-Mali. In these cases, a witness (other than study staff) also signed the

consent form. If a child arrived without a parent or guardian in attendance, they were not considered for enrollment in the study.

After enrollment, study staff collected demographic, historical and clinical information from the parent or guardian. All information was collected on a paper case report form (CRF). The 'present patient' clinical variables were entered into the App on a mobile device (Bangladesh: Samsung Galaxy A51; Mali: Samsung Galaxy Note 10) by two different study nurses independently to ensure reliability; variables were age in months for 0–23 months and in years for 2–4 years, blood in stool since illness began, history of vomiting (three or more times a day since illness started), breastfeeding status ('currently'), and mid-upper arm circumference (MUAC). During data collection, study investigators noted that nearly all participants in Bangladesh had reported 'yes' to the question regarding history of vomiting. Given vomiting was not expected in all patients especially those with non-viral diarrhea etiology, it was determined after speaking with the study nurses that the phrasing of the question sometimes led patients to respond 'yes' if there had been regurgitation with feeding or ORS administration, rather than actual vomiting. The question format was then revised for the remainder of the study enrollment in Bangladesh, and prior to any patient enrollment in Mali, to clarify the definition of vomiting. The App calculated the probabilities specific to each data source (present patient, recent patient, historical patient, climate, and viral seasonality). The model deployed on the App used the present patient and location-specific viral seasonality components along with the location-specific viral diarrhea alone prevalence from GEMS as pre-test odds.

The App results were not used for clinical decision-making to allow first for the external validation and second to iterate the software in response to engineering challenges exposed from 'live deployment'. All patients were treated according to standard local clinical protocols, and the clinicians caring for patients were blinded to any study data collected in order to prevent any undue influence in clinical care. Study procedures were not allowed to delay any immediately necessary patient care, such as the placement of an intravenous line or delivery of intravenous fluid to the patient. As the primary outcome was determined only after all clinical enrollment and procedures concluded and laboratory analysis was not linked to predictors, all assessments of predictors and the outcome were blinded.

## Sample collection and laboratory procedures

The first available stool specimen after enrollment of the participant was collected. Study participation concluded after a stool sample was obtained. Nurses were unaware of the etiology of diarrhea at the time of clinical assessment as microbiological testing was conducted only after the study period concluded. Stool samples were collected in a sterile plastic container and then transferred to two separated 2 mL cryovials – one vial with 1 mL stool only and one vial for storage in 70% ethanol (Bangladesh) or 95% ethanol (Mali). Samples were stored at –20°C or –80°C freezer for processing. At the conclusion of the study, samples were thawed, underwent bead beating and nucleic acid extraction using the QIAamp Fast DNA Stool Mini Kit. Total Nucleic acid was mixed with PCR buffer and enzyme and loaded onto custom multiplex TaqMan Array Cards (TAC) containing compartmentalized probe-based quantitative real-time PCR (qPCR) assays for 32 pathogens at the icddr,b or CVD-Mali laboratories (see *Supplementary file 2* for full list of pathogen targets).

## Assignment of diarrhea etiology

The outcome (dependent) variable was defined as the presence or absence of viral-only etiology. Diarrheal etiology was determined for each patient using qPCR attribution models developed previously by *Liu et al., 2016* Viral-only diarrhea was defined as a diarrhea episode with at least one viral pathogen with an episode-specific attributable fraction (AFe) threshold of ≥0.5 and no bacterial or parasitic pathogens with an AFe ≥0.5. This clinically relevant outcome measure was selected because patients with viral-only diarrhea should not receive antibiotics. Other etiologies were defined as having a majority attribution of diarrhea episode by at least one other non-viral pathogen. Patients without an attributable pathogen (unknown final etiology for diarrheal episode) were excluded from this analysis since the cause of the diarrheal episode could not be definitively determined. However, prior studies by this research team have shown that these cases have a similar distribution of viral predictions using a model with presenting patient information as those cases with known etiologies and this study had similar results *Brintz et al., 2020*.

## Data analysis

For the primary analysis, MUAC measurements collected by the two study nurses were averaged. Patients were considered to have 'bloody stool' only if report from both nurses agreed on bloody stool. For children older than 2 years, age in months was rounded down to the nearest year in months (i.e. 42 months was rounded to 36 months) to match the user interface on the software. Using clinical information gathered from the data sources (present patient, recent patient, historical patient, climate, viral seasonality), predictions using post-test odds formulation with the developed models were made. The primary model deployed in the App, selected based on the best-performing model from the derivation and internal validation, used the present patient data and viral seasonality components.

Model performance for the prediction of viral-only diarrhea was calculated using AUC for each model to evaluate discrimination; calibration was assessed using calibration-in-the-large and calibration slope *Steyerberg and Vergouwe, 2014*. The target for calibration slope is 1, where < 1 suggests predictions are too extreme and >1 suggests predictions are too moderate. The target for calibration intercept is 0, where negative values suggest overestimation and positive values suggest underestimation *Van Calster et al., 2019*. We estimated the calibration coefficients by regressing predicted values versus the observed proportion of viral cases, calculated using the observed proportion of viral cases within 0.05 plus or minus the predicted probability. For the primary analysis, data from the time period using the original vomiting question in Bangladesh was excluded; however, site specific results incorporate all Bangladesh data for the purpose of highlighting the potential impact of misunderstanding of the required App input questions in real-world scenarios. The reliability of the predictor data entered independently by the two study nurses was assessed using Cohen's kappa coefficient (κ) which is a calculation of inter-observer agreement for categorical data.

## Secondary analysis

The performance of additional models integrating other available components was assessed in post-hoc secondary analysis for comparison. These alternate models included the following components: (1) 'present patient' data only (2) 'present patient'+ climate data (climate) (3) 'present patient'+ historical prevalence pre-test odds (historical) (4) 'present patient'+ recent patient pre-test odds (recent) as described in *Table 1*.

For all analyses, a two-tailed p value of 0.05 was considered statistically significant. R (R Foundation for Statistical Computing, Vienna, Austria) were used for all analyses. Standard guidelines from the transparent reporting of a multivariable prediction model for individual diagnosis (TRIPOD) Checklist for Prediction Model Validation were used. The de-identified dataset is available online in *Supplementary file 5*.

## Sample size calculation

The previously derived clinical prediction model had an internally validated AUC of 0.83 *Brintz et al., 2021*. In order to ensure that the confidence intervals around the estimates in the validation study would not cross 0.75 (generally considered to be a marker of adequate accuracy for a clinical prediction model) a margin of error around the AUC of 0.08 was required. Using the approximate variance estimate for AUC from the literature, and assuming a prevalence of a viral only etiology of approximately 30% in our sample, a sample size of 300 patients (150 patients per site) was required.

## Results

### Participant characteristics

A total of 302 patients were recruited from the two study sites with 152 patients in Bangladesh and 150 patients in Mali. All patients except two in Bangladesh had a stool sample collected for TAC testing, for a total of 300 patients with TAC results (*Figure 2*). Diarrhea etiology was assigned for a total of 199 patients (66%; 130 in Bangladesh and 69 in Mali) for inclusion in the final analysis (*Table 2*). The median [IQR] age of included patients was 12 months; there was a predominance of male patients at both study sites (61.8% overall) with 59.2% in Bangladesh and 66.7% in Mali. Socio-demographic and clinical characteristics of the study participants are shown in *Table 2*. In Bangladesh, 73 participants were asked about vomiting history using the original question format and 57 using the revised question format. Antibiotic use prior to health facility presentation was common with 66%

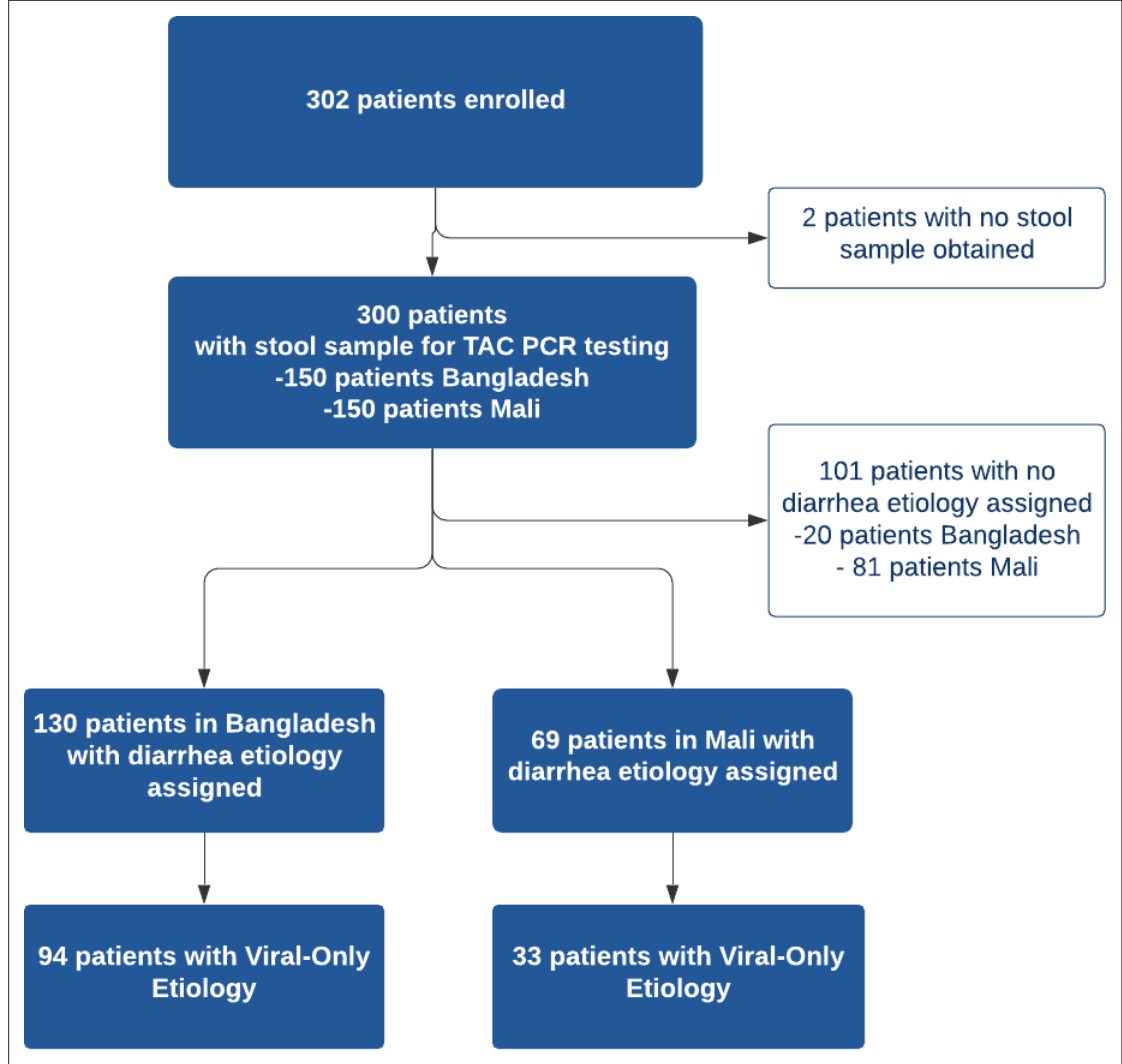

**Figure 2.** Study Flow Diagram.

of participants in Bangladesh reporting antibiotic use for the current illness (most commonly azithromycin, ciprofloxacin, and metronidazole alone or in combination) and 13.1% of participants Mali (most commonly cotrimoxazole, amoxicillin, metronidazole).

## Diarrhea etiology

Viral-only etiologies of diarrhea were the predominant cause of illness among patients with assigned etiology overall (42%) although viral-only etiologies were more common in Bangladesh (63%) compared to Mali (22.0%). Rotavirus was the most common viral pathogen detected in both study sites (38% overall) while the rate was higher in Bangladesh (60%) compared to Mali (16%). Other viral causes (e.g. adenovirus, astrovirus, norovirus) were less common in both sites. Diarrhea etiology was unable to be assigned in a substantial proportion of the patients in Mali (54%) compared to Bangladesh (13.3%). The prevalence of the various pathogens detected are listed in *Table 3*.

## Clinical prediction models performance: primary and secondary models

When applied to both sites, the primary model which included present patient+ viral seasonality component performed better than secondary models at discriminating a viral-only etiology from all other known etiologies with an AUC of 0.754 (95% CI 0.665–0.843) (*Table 4*). However, this same model was not well calibrated with an estimate of calibration-in-the-large of $\alpha$ = −0.393 (95% CI -0.455–-0.331) and a calibration slope of $\beta$ = 1.287 (95% CI 1.207, 1.367). The present patient+

**Table 2.** Clinical characteristics of study population.

| | Diarrhea etiology assigned | | | No diarrhea etiology assigned | | |
|---|---|---|---|---|---|---|
| | Overall n(%) N = 199 | Bangladesh n(%) N = 130 | Mali n(%) N = 69 | Overall n(%) N = 101 | Bangladesh n(%) N = 20 | Mali n(%) N = 81 |
| Age (median, IQR), months | 12 (8) | 12 (9) | 11 (8) | 8 (7) | 11.5 (7) | 8 (7) |
| Sex | | | | | | |
| Male | 123 (61.8) | 77 (59.2) | 46 (66.7) | 63 (62.4) | 11 (55) | 52 (64.2) |
| Female | 76 (38.2) | 53 (40.8) | 23 (33.3) | 38 (37.6) | 9 (45) | 29 (35.8) |
| Diarrhea Duration (median, IQR), days | 2 (2) | 3 (1) | 0.6 (0.3) | 1.5 (2.3) | 3 (1.3) | 0.7 (0.1) |
| # Episodes of Diarrhea Past 24 hours (median, IQR) | 12 (9.5) | 15 (8) | 5 (3) | 6 (4) | 15 (3.5) | 5 (3) |
| Bloody Stool Reported | | | | | | |
| Yes | 7 (3.5) | 2 (1.5) | 5 (7.2) | 3 (3) | 0 (0) | 3 (3.7) |
| No | 192 (96.5) | 128 (98.5) | 64 (92.8) | 98 (97) | 20 (100) | 78 (96.3) |
| Fever Reported | | | | | | |
| Yes | 163 (81.9) | 119 (91.5) | 44 (63.8) | 66 (65.3) | 18 (90) | 48 (59.3) |
| No | 36 (18.1) | 11 (8.5) | 25 (36.2) | 35 (34.7) | 2 (10) | 33 (40.7) |
| Vomiting Reported** | | | | | | |
| Yes (original question format) | 73 (36.7) | 73 (56.2) | | 12 (11.9) | 12 (60) | |
| Yes (revised question format) | 81 (40.7) | 45 (34.6) | 36 (52.2) | 21 (20.8) | 5 (25) | 16 (19.8) |
| No | 45 (22.6) | 12 (9.2) | 33 (47.8) | 68 (67.3) | 3 (15) | 65 (80.2) |
| Breastfeeding | | | | | | |
| Yes (Partial or Exclusive) | 155 (77.9) | 99 (76.2) | 56 (81.2) | 89 (88.1) | 19 (95) | 70 (86.4) |
| No | 44 (22.1) | 31 (23.8) | 13 (18.8) | 12 (11.9) | 1 (5) | 11 (13.6) |
| MUAC (median, IQR), cm | 13.55 (1.4) | 13.45 (1.3) | 13.8 (1.5) | 13.35 (1.8) | 13.375 (2.2) | 13.35 (1.9) |
| Prior Medications Taken | | | | | | |
| Yes | 124 (62.3) | 109 (83.8) | 15 (21.7) | 34 (33.7) | 17 (85) | 17 (21) |
| No | 75 (37.7) | 21 (16.2) | 54 (78.3) | 67 (66.3) | 3 (15) | 64 (79) |
| Years of Mother's Education | 8 (6) | 8 (5) | 4 (9) | 6 (10) | 8 (4.75) | 5 (10) |
| Years of Father's Education | 8 (6) | 8 (5) | 6 (9) | 6 (10) | 10 (4.8) | 4 (10) |
| People living at home (median, IQR) | 6 (4) | 5 (2) | 9 (12) | 9 (10) | 6 (6) | 10 (9) |

*Abbreviations: IQR, interquartile range; cm, centimeter; MUAC, mid-upper arm circumference*
*** Vomiting question was asked in two different formats at the Bangladesh site*

**Table 3.** Pathogens detected with TaqMan array card by study site.

| Bangladesh N = **150** | | Mali N = **150** | |
|---|---|---|---|
| Pathogen | n (%) | Pathogen | n (%) |
| No Etiology Assigned | 20 (13) | No Etiology Assigned | 81 (54) |
| *Viral-Only Pathogens* | 94 (63) | *Viral-Only Pathogens* | 33 (22) |
| Rotavirus | 90 (60) | Rotavirus | 24 (16) |
| Adenovirus 40/41 | 1 (0.6) | Norovirus GII | 5 (3.3) |
| Astrovirus | 2 (1.3) | Astrovirus | 2 (1.3) |
| Adenovirus & Astrovirus | 1 (0.6) | Astrovirus & Rotavirus | 2 (1.3) |
| *Bacterial-Only Pathogens* | 9 (6) | *Bacterial-Only Pathogens* | 24 (16) |
| *Vibrio cholerae* | 3 (2) | *Shigella* / EIEC | 13 (8.7) |
| *Shigella / Enteroinvasive E. coli (EIEC)* | 1 (0.6) | Shiga-toxin Enterotoxigenic *E. coli* | 4 (2.7) |
| *Campylobacter jejuni / coli* | 1 (0.6) | *Campylobacter jejuni / coli* | 2 (1.3) |
| Multiple Bacterial Pathogens | 4 (2.7) | *Salmonella* | 2 (1.3) |
| | | Multiple Bacterial Pathogens | 3 (2) |
| *Parasitic Pathogens* | 1 (0.6) | *Parasitic Pathogens* | 5 (3.3) |
| Cryptosporidium | 1 (0.6) | Cryptosporidium | 4 (2.7) |
| | | *Entamoeba histolytica* | 1 (0.6) |
| *Mixed Pathogens* | 26 (17) | *Mixed Pathogens* | 7 (4.7) |
| Viral+ Bacteria | 24 (16) | Viral+ Bacteria | 7 (4.7) |
| Viral+ Bacterial + Parasitic | 2 (1.3) | | |

historical patient model having low discrimination AUC 0.702 (95%CI 0.603–0.800) was the best calibrated with estimates of 0.036 (−0.031–0.102) and 1.063 (0.943–1.184) for α and β, respectively. Combinations of more than two components including the present patient information reduced the discriminatory performance of the model, likely due the positive correlation between components.

When looking at the sites individually, the present patient+ recent patient model performed best in Mali with an AUC of 0.783 (95% CI 0.705–0.86) while the primary model with the present patient+ viral seasonality component data performed best in Bangladesh with an AUC of 0.71 (95% CI 0.595–0.825) (*Table 5*). When the dates where the vomiting question was asked incorrectly are included in the analysis, all models performed less well with the viral seasonality and climate models performing best in Bangladesh with an AUC of 0.610 (95% CI: 0.523–0.697) and 0.621 (95% CI: 0.510–0.732), respectively (*Table 5*).

While the aim of the study was not to provide a binary classification decision but rather a continuous predicted risk of viral etiology, the numbers of false positives and false negatives (i.e, bacteria/

**Table 4.** Model performance using AUC (95% confidence interval), calibration-in-the-large (α), calibration slope (β) for each model considered at both sites.
Each row after 'Present patient' includes the Present patient component.

| | Auc (95% CI**)** | α | β |
|---|---|---|---|
| Present patient | 0.744 (0.651–0.836) | −0.212 (−0.264–−0.16) | 1.250 (1.171–1.329) |
| Viral seasonality | 0.754 (0.665–0.843) | −0.393 (−0.455–0.331) | 1.287 (1.207–1.367) |
| Climate | 0.680 (0.583–0.778) | −0.115 (−0.191–−0.038) | 0.940 (0.840–1.039) |
| Historical patient | 0.702 (0.603–0.800) | 0.036 (−0.031–0.102) | 1.063 (0.943–1.184) |
| Recent patient | 0.737 (0.671–0.793) | −0.253 (−0.287–−0.22) | 1.165 (1.12–1.21) |

**Table 5.** AUC (95% confidence interval) for each model by site.
Each row after 'Present patient' includes the Present patient component. The last column includes Bangladesh data in which the vomiting question was asked incorrectly.

|  | Mali | Bangladesh | Bangladesh (no date restriction) |
|---|---|---|---|
| Present patient | 0.763 (0.681–0.844) | 0.692 (0.572–0.812) | 0.607 (0.521–0.693) |
| Viral seasonality | 0.742 (0.659–0.825) | 0.71 (0.595–0.825) | 0.61 (0.523–0.697) |
| Climate | 0.701 (0.577–0.825) | 0.607 (0.427–0.788) | 0.621 (0.510–0.732) |
| Historical patient | 0.741 (0.658–0.824) | 0.646 (0.516–0.775) | 0.592 (0.505–0.86) |
| Recent patient | 0.783 (0.705–0.86) | 0.625 (0.5–0.75) | 0.61 (0.526–0.694) |

protozoal etiologies misidentified as viral and vice versa) at various viral probability thresholds for 'present patient' and the 'present patient+ viral seasonality' models are shown in *Figure 3*. Notably, the 'present patient+ viral seasonality' model tended to have fewer false negatives while the 'present patient' model fewer false positives at various thresholds. However, the sensitivity and specificity of the 'present patient' and 'present patient+ viral seasonality' models were similar (*Figure 3*).

### Reliability of software use

The agreement between the two study nurses' independent assessments of the required predictor variables and estimated viral-only etiology risk calculated by the App for each nurse's assessment was evaluated and showed excellent reliability and agreement between the nurses; results are shown in *Supplementary file 3*. Data from the App were recorded through the cellular network to a database as well as on the paper CRF. To monitor integrity of the data transfer to the App database from a technical perspective, data were compared between the two records. The agreement between the App database and study nurse's paper CRF are shown in *Supplementary file 4* and *Figure 4—figure supplement 1*. The Bangladesh study exposed that one field ('breastfeeding') did not sync successfully which was addressed at the midpoint between the Bangladesh and Mali study phases. The Mali study used an updated software version (v2.2.5); the reliability and agreement between the App prediction of viral etiology with the post-hoc predictions are shown in *Figure 4*.

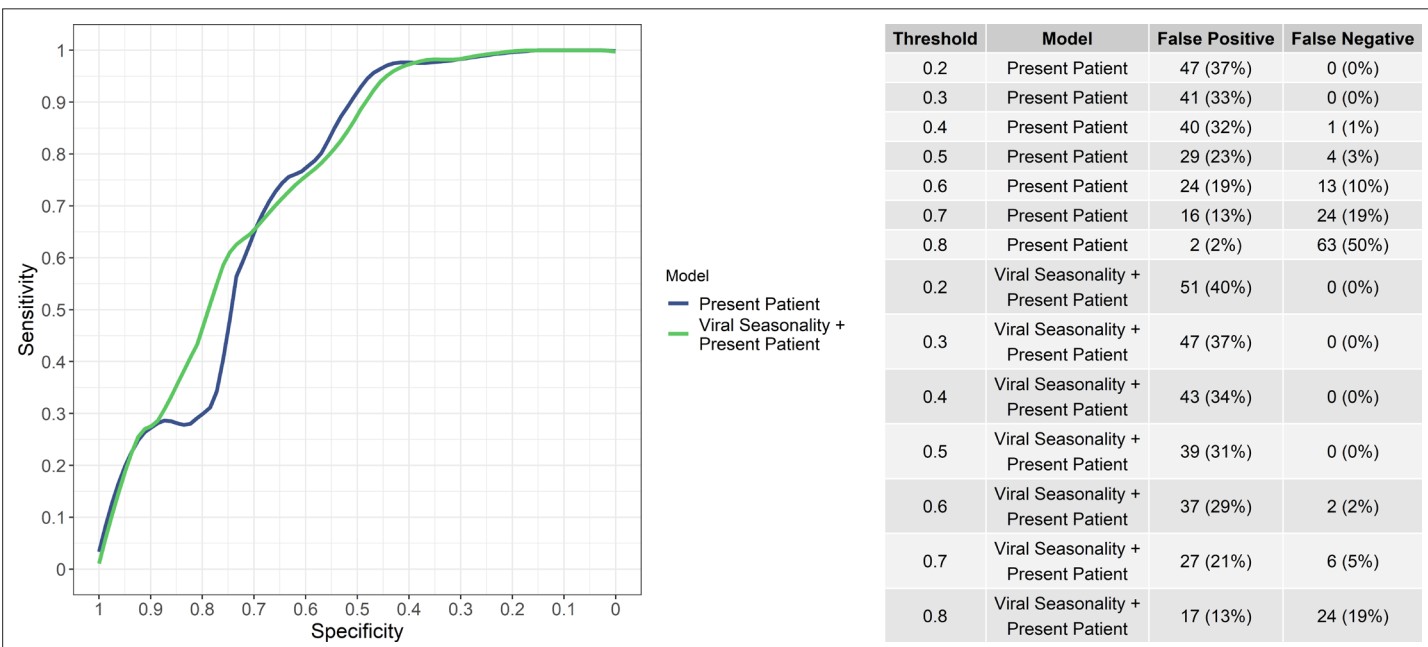

| Threshold | Model | False Positive | False Negative |
|---|---|---|---|
| 0.2 | Present Patient | 47 (37%) | 0 (0%) |
| 0.3 | Present Patient | 41 (33%) | 0 (0%) |
| 0.4 | Present Patient | 40 (32%) | 1 (1%) |
| 0.5 | Present Patient | 29 (23%) | 4 (3%) |
| 0.6 | Present Patient | 24 (19%) | 13 (10%) |
| 0.7 | Present Patient | 16 (13%) | 24 (19%) |
| 0.8 | Present Patient | 2 (2%) | 63 (50%) |
| 0.2 | Viral Seasonality + Present Patient | 51 (40%) | 0 (0%) |
| 0.3 | Viral Seasonality + Present Patient | 47 (37%) | 0 (0%) |
| 0.4 | Viral Seasonality + Present Patient | 43 (34%) | 0 (0%) |
| 0.5 | Viral Seasonality + Present Patient | 39 (31%) | 0 (0%) |
| 0.6 | Viral Seasonality + Present Patient | 37 (29%) | 2 (2%) |
| 0.7 | Viral Seasonality + Present Patient | 27 (21%) | 6 (5%) |
| 0.8 | Viral Seasonality + Present Patient | 17 (13%) | 24 (19%) |

**Figure 3.** Sensitivity and specificity of the 'present patient' and 'present patient+ viral seasonality' models (left) and numbers of false positives and false negatives (i.e, bacteria/protozoal etiologies misidentified as viral and vice versa) at various viral probability thresholds (right).

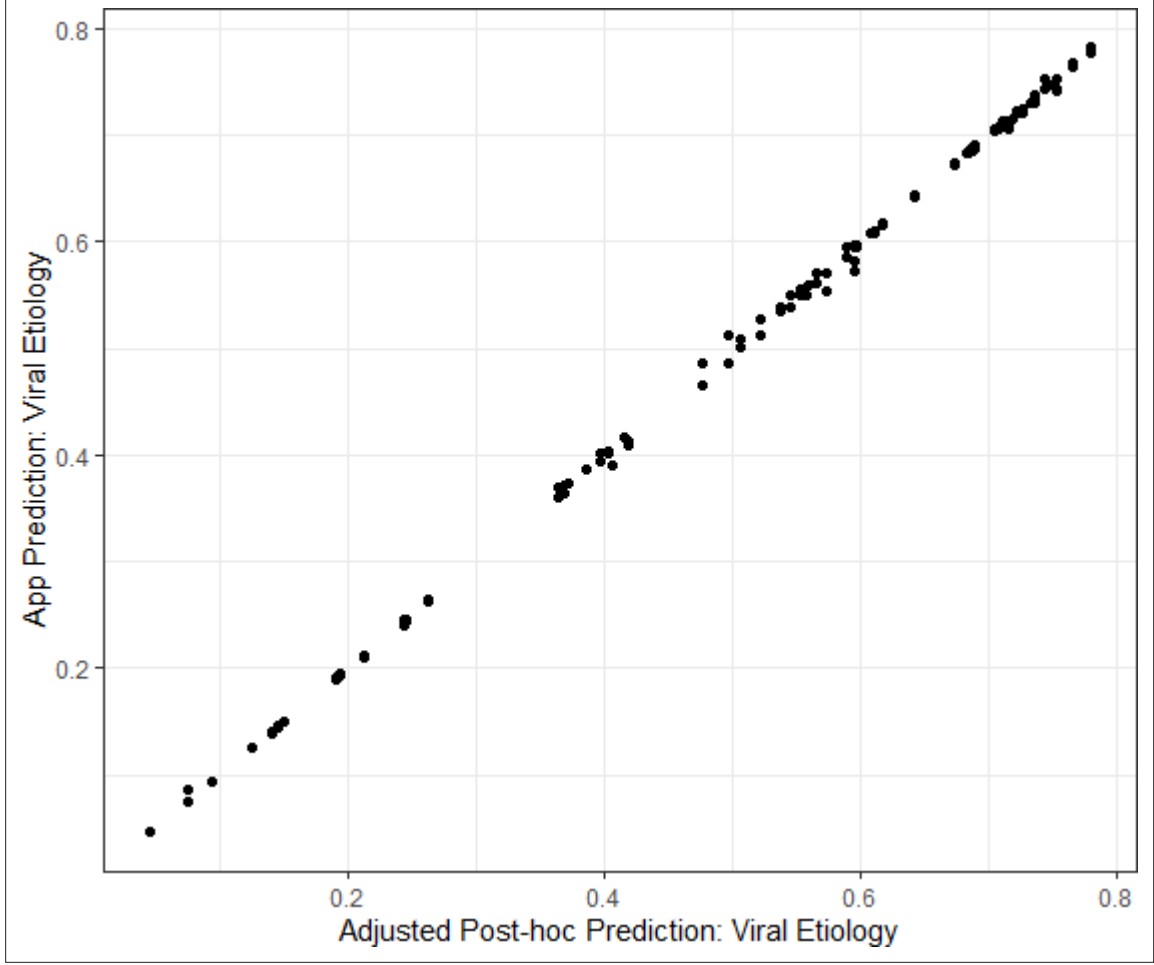

**Figure 4.** Congruence between the Application prediction of a patient's viral etiology with the post-hoc prediction after adjusting for changing model development. Data shown are from the Mali study period alone because the software was updated between the Bangladesh and Mali study periods in response to engineering limitations.

The online version of this article includes the following figure supplement(s) for figure 4:

**Figure supplement 1.** Bland-Altman plots showing agreement between data entered on case report form versus App for mid-upper arm circumference and calculated predicted viral-only etiology risk.

## Discussion

This prospective, clinical study provides an external validation of a mobile CDSS for the prediction of diarrhea etiology in children under five years old. Combining the strengths of prediction models generated through a Bayesian approach, the novelty of etiological prediction, and the accessibility gained by incorporation into software on a mobile device, this study represents an important advancement in translational research to build evidence-based tools to improving care in resource-limited settings. Incorporating the internally and now externally validated etiology prediction tool as a component of CDSS for diarrheal diseases could have broad scalable impact, including the potential for reducing inappropriate antibiotic use.

Despite recommendations from the WHO discouraging antibiotic use for the majority of presentations of pediatric diarrhea, multiple studies from LMICs have shown that antibiotic overuse and inappropriate antibiotic prescribing remains widespread with up to 50–80% of children with diarrhea under five receiving antibiotics *Bebell and Muiru, 2014*; *Ahmed et al., 2009*; *De et al., 2016*; *Fink et al., 2020*; *Tsige et al., 2020*. While antibiotic use is widespread in HICs and LMICs, antibiotic overuse is particularly challenging in LMICs for a multitude of reasons including: widespread availability and unregulated sale of antibiotics, long-standing patient expectations for antibiotics, low

public health knowledge of AMR, and limited diagnostic testing capacity *Bebell and Muiru, 2014*; *Vila and Pal, 2010*; *Okeke et al., 2005*; *World Health Organization, 2017*; *Klein et al., 2018*. A 2020 re-analysis of results from the GEMS study also estimated that there were approximately 12.6 inappropriate-treated diarrhea cases for each appropriately-treated case, with viruses (rotavirus, adenovirus, sapovirus) among the leading antibiotic-treated etiologies *Lewnard et al., 2020*. Tools such as the App created for this study may provide clinicians with much-needed evidence to better identify patients with viral-only diarrhea and increase their confidence in following guidelines when antibiotic use is not indicated.

Used for a broad variety of applications in HICs, CDSSs have been beneficial for patient and health systems outcomes alike; CDSSs have been shown to improve antibiotic stewardship and reduce inappropriate prescription of antibiotics, reduce the use of high-cost computed tomography (CT) studies in emergency department patients, and improve glucose control in patients with diabetes, among many other examples *Sintchenko et al., 2008*; *Bookman et al., 2017*; *Carracedo-Martinez et al., 2019*. In the United States, a randomized controlled trial in rural communities in Utah and Idaho showed that an antibiotic stewardship CDSS run on a mobile device reduced overall antimicrobial use and improved appropriateness of antibiotic selection for respiratory tract infections *Samore et al., 2005*. While a lack of research persists regarding CDSSs in LMICs, their use has risen recently due to the increasingly ubiquitous use of smartphones globally. Mobile CDSSs have been successfully used to improve hypertension control in India, and have been developed for neonatal care in Ghana, Kenya, and Uganda *Watson et al., 2019*; *Anchala et al., 2015*; *Muhindo et al., 2021*; *Bucher et al., 2021*; *Amoakoh et al., 2017*; *Bilal et al., 2018*. In Bangladesh, electronic CDSSs have also been shown to improve WHO diarrhea guideline adherence including reducing non-indicated antibiotic use by 28.5% in children under five *Khan et al., 2020*.

While prior studies of CDSS targeting antibiotic use have focused on improving adherence to guidelines, our prediction tool provides the clinician with an estimation of viral diarrhea etiology. Such information recognizes the expertise of the end-user and augments the available information that is used in the decision-making process for antibiotic use, potentially improving trust in the tool. In addition, entry of a select few clinical variables that can be easily collected at the point-of-care into the prediction tool may enhance clinician awareness of the relevance of such factors and further increase confidence in the use of the tool. A 2021 pre-implementation workshop among health professionals in Burkina Faso aimed at developing a CDSS for appropriate antimicrobial use in West Africa found that although the tool was expected to reduce antibiotic use, the lack of epidemiologic and microbiological data and limited availability of diagnostic tests were cited as anticipated barriers to use *Peiffer-Smadja et al., 2020*. Our App provides this additional information taking local epidemiologic data into account which may increase provider buy-in toward antibiotic stewardship efforts. Our study also provides an approach by which similar models could be developed for other infectious syndromes such as respiratory and febrile illness.

A unique aspect of this study was the use of a modular approach to clinical prediction models. In this validation study, the App deployed a primary model of present patient+ viral seasonality data based on the best performing model in internal validation studies. However, the great strength of the modular approach is that it allows for customization according to data availability and local diarrhea epidemiology. In this validation study, we found that the model performance varied depending on the study site. To that end, when possible, a trial period would be prudent prior to implementation of the App to determine which model has the best performance for the selected site. In the present study, we found that the best performing models at each site had a discriminatory performance, as assessed by AUC, of 0.783 (0.705–0.86) in Mali and 0.710 (0.595–0.825) in Bangladesh. These levels of performance are comparable to that achieved in external validation of one of the most commonly-used prediction models for etiology of infection, the Centor score for streptococcal pharyngitis, which achieved an AUC of 0.72 in two separate external validation studies *Wigton et al., 1986*; *Fine et al., 2012*. Current WHO IMCI guidelines recommend use of antibiotics only for cases with bloody diarrhea or suspicion for cholera with severe dehydration, which account for less than 10% of children with diarrhea worldwide *Nelson et al., 2011*; *World Health Organization, 2005b*. In the majority of cases where antibiotics are not indicated, even a prediction rule with moderate performance would help to reduce inappropriate antibiotic use. While setting a high viral probability threshold would ensure that non-viral cases are not missed, such a threshold may not be as important

if the implementation of this prediction rule is limited to children who do not qualify for antibiotics based on WHO recommendations.

Importantly, all data collection for the App was conducted by nurses, the majority of whom had never used a mobile CDSS for clinical use previously. Nurses were able to confidently use the smartphone after a relatively brief training period and enter all necessary data into the tool without difficulty in high-volume clinical settings. While the ease of use or exact time required to enter data into the App was not formally assessed in this validation study, from direct observation nurses were able to enter all required data and obtained calculated outputs within several minutes per patient. Given this time required is similar to that spent with patients during routine clinical care, we believe the App would have high clinical utility in the busy clinical settings for which it was designed. Although the calculations for the primary model deployed on the app were performed on-device, data was transmitted successfully from the App to the Cloud server, from an engineering standpoint this indicates that using real-time data sources is feasible. Additionally, the excellent reliability and agreement between nurses as well as between the App and the paper CRF reference demonstrate that accurate data collection is highly feasible in these clinical settings. Formal evaluation of the usability of the App including an assessment of user experience and metrics such as time required for data entry and for display of calculated outputs using real-time data sources are also needed.

Lastly, given the substantial drop in AUC when the vomiting question was asked incorrectly, a trial period to ensure precise understanding of the App's questions among the target population is recommended. These findings underscore the challenges of implementing mHealth tools in diverse contexts, and the importance of ensuring cross-cultural understanding of questions by clinician and patient end-users in real-world scenarios.

## Limitations

Our findings have several limitations. The study was conducted at study sites located in close proximity to sites from which the prediction model data were trained on. The prediction models may not therefore be generalizable to other locations, sites that have poor epidemiologic data on diarrhea etiologies, or locations that have initiated new vaccination campaigns (e.g. rotavirus). Rotavirus vaccination has not yet been introduced in Bangladesh, although a cluster randomized trial was conducted in rural Matlab in 2008–2011, and rotavirus vaccine introduction into the routine immunization program has been planned; thus, we do not expect any effect of the vaccine on the present study results for the Bangladesh study site. However, in Mali, rotavirus vaccination was introduced into the routine immunization program in 2014, as of 2019 official coverage rate was 63% in infants 12–23 months *Zaman et al., 2017*; *Rota Council, 2021*; *Mali, 2019*. We recognize that the App may not perform as well in places where the rotavirus vaccine has been introduced given the change in viral diarrhea epidemiology in the years since the data were collected from which the models were trained. However, there remain many countries without access to the rotavirus vaccine, notably in Asia where only a few countries have introduced the vaccine nationally or sub-nationally *Rota Council, 2021*. For this reason, the App may be most useful in these settings. Interestingly, the models performed better at the Mali study site compared to Bangladesh despite fairly high coverage rates of rotavirus vaccine with subsequently much lower rates of rotavirus detected on TAC PCR. These findings may indicate that the clinical features associated with viral diarrhea may be pathogen-independent, and that seasonal variations in viral diarrhea may still be relevant even when overall rates of rotavirus are substantially lower than in prior time periods. Lastly, as the rotavirus vaccine is planned to be introduced in an increasing number of countries globally, we also recognize that the model may need to be updated to account for changes in epidemiology as rotavirus vaccine is increasingly introduced and rotavirus becomes a less prominent cause of viral diarrhea.

Another limitation is that approximately one-third of patients enrolled in this study did not have diarrhea etiology assigned using the predefined threshold AFe >0.5 from the TAC testing data. The much higher rate of diarrhea etiology assignment in Bangladesh compared to Mali may be partially attributable to the study being conducted during the annual rotavirus season in Bangladesh (typically November to February each year) with rotavirus strongly associated with diarrhea and therefore easier to attribute *Dey et al., 2020*. However, the predominance of rotavirus may limit the generalizability of this study's findings. Additionally, given the known seasonal cycles of viral diarrhea infections, ideally, at least one full annual cycle of data would have been collected at each study site to account

for seasonal variations in diarrhea epidemiology. However, funding and resource limitations, and the prioritization of including two diverse study sites precluded a longer duration of study enrollment. Notably, the occurrence of a large seasonal rotavirus peak during the data collection period in Bangladesh may be a reason why the models performed sub-optimally at this site in comparison to Mali. We believe that, had resources been available for a longer duration of study enrollment that accounted for greater seasonal variability, our models would have shown better performance characteristics than in the present analysis. Studies to further validate these models using additional study sites with different seasonal cycles of viral diarrhea over at minimum one full annual cycle are recommended. Changes over time in the epidemiology of diarrhea at the study sites as well as the internal validation being conducted in a very large dataset ( > 20,000 patients) from the GEMS study compared to this study with 300 patients over a short period during a surge of rotavirus likely also contributed to the drop in AUC between the internal validation and external validation.

Lastly, there is currently no existing gold standard for threshold of AFe that should be used to assign etiology and the threshold of 0.5 was set a priori based on expert consultation; however, the effect of using this cut-off has not yet been explored. A lower AFe threshold may still indicate likely etiology of diarrhea. Notably, there were no norovirus infections detected using these pre-determined thresholds. This may have been due to the particularly high prevalence of other pathogens especially rotavirus in Bangladesh during this time period, the previously documented lower prevalence of norovirus in Bangladesh in November-January compared to other months, as well as the relatively short period of participant enrollment, although using a different AFe cutoff may have detected possible cases of norovirus *Satter et al., 2021* .

## Conclusion

Despite these limitations, we have externally validated, in a prospective multicenter study, a smartphone-based clinical decision-support system that dynamically implements clinical prediction models based on real-time data streams. A clinical trial led by this study team is currently underway to evaluate the impact of this tool on antibiotic prescribing behaviors, and further research is needed to better understand how these tools could be best adapted for use by practicing clinicians in busy LMIC settings.

## Acknowledgements

The authors thank all study participants and the study staff of the icddr,b Dhaka Hospital and the health centers of Commune V and VI in Mali for their help and support. The authors also thank the development team at BeeHyv Software Solutions Pvt. Ltd. (Wilmington, DE; Hyderabad, India) who were instrumental in building the digital clinical decision support software used in this study. Funding for this study was provided through grants from the Bill and Melinda Gates Foundation (OPP1198876) and the National Institute of Allergy and Infectious Diseases (R01AI135114). Several investigators were also partially supported by a grant from the National Institute of Diabetes and Digestive and Kidney Diseases (R01DK116163). This investigation was also supported by the University of Utah Population Health Research (PHR) Foundation, with funding in part from the National Center for Advancing Translational Sciences of the National Institutes of Health under Award Number UL1TR002538. The software architecture from which the App was constructed was supported by the National Institutes of Health [DP5OD019893; R21TW010182] to EJN. The content is solely the responsibility of the authors and does not necessarily represent the official views of the National Institutes of Health. The funders had no role in the study design, data collection, data analysis, interpretation of data, or in the writing or decision to submit the manuscript for publication.

## Additional information

### Competing interests

Stephanie Chow Garbern: received consultancy fees from the University of Utah, in relation to project "Development of clinical decision tools for management of diarrhea of children in high and low resource settings" (R01AI135114, PI: Leung), and from Department of Defense Naval Medical Logistics

Command, for the Austere Environments Consortium for Enhanced Sepsis Outcomes: An Observational Study of Sepsis (N62645-14-2-0001). The author has no other competing interests to declare. Eric J Nelson: is associated with a patent on data collection components in the 'Outbreak Responder' (Patent Publication Number 2020/0082921), of which the original 'Rehydration Calculator' was a component. These components are not included in the software referenced and published herein. Has no financial interest in the 'Rehydration Calculator' herein or the original 'Outbreak Responder' software. Dilruba Nasrin: received travel funding for training and monitoring data collection in Mali. The author has no other competing interests to declare. Adam C Levine: received consultancy fees from the University of Utah, in relation to project "Development of clinical decision tools for management of diarrhea of children in high and low resource settings" (R01AI135114, PI: Leung). The author has no other competing interests to declare. The other authors declare that no competing interests exist.

## Funding

| Funder | Grant reference number | Author |
|---|---|---|
| Bill and Melinda Gates Foundation | OPP1198876 | Daniel T Leung |
| National Institute of Allergy and Infectious Diseases | R01AI135114 | Daniel T Leung |
| National Institute of Diabetes and Digestive and Kidney Diseases | R01DK116163 | Adam C Levine Monique Gainey |
| National Center for Advancing Translational Sciences | UL1TR002538 | Ben J Brintz |
| National Institutes of Health | R21TW010182 | Eric J Nelson |

The funders had no role in study design, data collection and interpretation, or the decision to submit the work for publication.

## Author contributions

Stephanie Chow Garbern, Data curation, Formal analysis, Investigation, Methodology, Project administration, Software, Supervision, Validation, Visualization, Writing – original draft, Writing – review and editing; Eric J Nelson, Conceptualization, Data curation, Formal analysis, Investigation, Methodology, Project administration, Resources, Software, Supervision, Validation, Visualization, Writing – original draft, Writing – review and editing; Sabiha Nasrin, Data curation, Investigation, Project administration, Writing – review and editing; Adama Mamby Keita, Data curation, Investigation, Project administration, Supervision, Writing – review and editing; Ben J Brintz, Data curation, Formal analysis, Methodology, Software, Validation, Visualization, Writing – original draft, Writing – review and editing; Monique Gainey, Investigation, Project administration, Writing – original draft, Writing – review and editing; Henry Badji, Investigation, Project administration, Writing – review and editing; Dilruba Nasrin, Mami Taniuchi, Karen L Kotloff, Resources, Writing – review and editing; Joel Howard, Conceptualization, Writing – review and editing; James A Platts-Mills, Conceptualization, Methodology, Resources, Writing – review and editing; Rashidul Haque, Investigation, Resources, Writing – review and editing; Adam C Levine, Conceptualization, Funding acquisition, Investigation, Methodology, Project administration, Resources, Supervision, Writing – review and editing; Samba O Sow, Conceptualization, Investigation, Project administration, Resources, Supervision, Writing – review and editing; Nur Haque Alam, Conceptualization, Investigation, Project administration, Resources, Supervision, Validation, Writing – review and editing; Daniel T Leung, Conceptualization, Data curation, Formal analysis, Funding acquisition, Investigation, Methodology, Project administration, Resources, Supervision, Validation, Visualization, Writing – original draft, Writing – review and editing

## Author ORCIDs

Stephanie Chow Garbern ⓘ http://orcid.org/0000-0002-0919-2841
Ben J Brintz ⓘ http://orcid.org/0000-0003-4695-0290
Monique Gainey ⓘ http://orcid.org/0000-0002-2860-9104
Daniel T Leung ⓘ http://orcid.org/0000-0001-8401-0801

## Ethics

Human subjects: Informed consent was obtained from all study participants as described in Materials and Methods. Ethical approval was obtained from the icddr,b Ethical Review Committee (PR-19095), the University of Sciences, Techniques, and Technologies of Bamako Ethics Committee (2019-153), and the University of Utah Institutional Review Board (IRB_00121790).

## Decision letter and Author response

Decision letter https://doi.org/10.7554/eLife.72294.sa1
Author response https://doi.org/10.7554/eLife.72294.sa2

---

# Additional files

## Supplementary files

• Supplementary file 1. Descriptive data – demographics, predictors and viral-only outcome data from development dataset from GEMS.

• Supplementary file 2. List of Pathogen Targets for Taqman Array Card Testing.

• Supplementary file 3. Assessment of reliability and agreement between study nurses' independent assessments of categorical predictor variables on case report forms.

• Supplementary file 4. Agreement between study nurse recording on paper case record and input into App of categorical predictor variables.

• Supplementary file 5. De-identified Dataset.

• Transparent reporting form

## Data availability

The de-identified dataset is included as Supplementary File 5 and has been deposited on Dryad. The modeling code and additional files needed to run the code are deposited on GitHub at https://github.com/LeungLab/DiaPR_Phase1, (copy archived at swh:1:rev:de136bd1a5d61cb2c98865277e292fbf76d18fff).

The following dataset was generated:

| Author(s) | Year | Dataset title | Dataset URL | Database and Identifier |
|---|---|---|---|---|
| Leung D | 2021 | Data from: Diarrhea Etiology Prediction Validation Dataset - Bangladesh and Mali | http://dx.doi.org/10.5061/dryad.0rxwdbs19 | Dryad Digital Repository, 10.5061/dryad.0rxwdbs19 |

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
