## [Decision Letter]

**Decision letter after peer review:**

Thank you for submitting your article "External Validation of a Mobile Clinical Decision Support System for Diarrhea Etiology Prediction in Children: A Multicenter Study in Bangladesh and Mali" for consideration by *eLife*. Your article has been reviewed by 2 peer reviewers, and the evaluation has been overseen by a Reviewing Editor and David Serwadda as the Senior Editor. The following individual involved in review of your submission has agreed to reveal their identity: Sanjat Kanjilal (Reviewer #1).

Essential revisions:

1) Please address the relative lack of seasonality in the data used for model validation in both cohorts.

2) Please discuss how operational components may impact the performance characteristics of the decision making tools.

3) Please discuss how frequently which the mode missed bacterial or protozoal infections.

4) Please limit jargon and provide definitions for all terms listed by Reviewer 2. The mechanistic and statistical explanations for α and β in the methods are also insufficient to interpret the results, particularly for the generalist reader. Please provide a lengthier and clearer explanation of the calibration procedure.

5) Please discuss the potential impact of the rotavirus vaccine on observed study outcomes.

*Reviewer #1:*

Garbern and Nelson et al., performed an external validation of clinical decision support system (CDSS) for prediction of a viral etiology for kids presenting with diarrhea in low and middle income countries, that is deployable on a mobile app. The prediction model is derived from data accrued in the Global Enteric Multicenter Study (GEMS), which completed in 2011 and defined etiologies and factors associated with pediatric diarrhea across 7 countries. The original GEMS samples were re-analyzed by a different group of investigators using a multiplex gastroenteritis syndromic panel to establish the 'ground truth' data upon which a random forest model was trained (Liu, Lancet, 2016). A simpler 5 feature logistic regression model was chosen using the variable weights assigned by the random forest and included only factors that would be obtainable by practitioners in real time. The final model utilized patient-intrinsic features as well as a viral seasonality model that they combined into a single prediction using a pre-determined cutoff (attributable fraction > 0.5) validated in previous work and by expert opinion. The study had a 'lead in' period where adjustments were made to the CDSS to adjust for real-world deployment and then they collected data from 2 referral centers, the first in Dhaka, Bangladesh and the second in Bamako, Mali.

The results of this external validation study show that model performance dropped relative to the internal validation (AUC 0.83) at both sites and were just at the arbitrary AUC cutoff of 0.75 for acceptable model accuracy, though with suboptimal calibration.

Strengths

Most antibiotic use in the setting of pediatric gastroenteritis is unnecessary. The impact of the CDSS for management of pediatric diarrhea in the literature appears mixed, with some studies showing increased antibiotic use and others showing decreased. This highlights the critical need for a rigorous prospective multicenter RCT to evaluate the utility of a point of care CDSS for optimal decision making around pediatric gastroenteritis. The results of the current study are an important stepping stone to that objective. The establishment of ground truth using a multiplex PCR assay provides for a more sensitive and (possibly) more specific reference by which to compare model outputs relative to traditional culture, EIA and uniplex PCR methods used in GEMS. Additionally, the use of a mobile app with a simple UI maximizes the number of staff members that will be able to utilize the CDSS. Finally, the use of an interpretable model that incorporates local data is an excellent design choice for improving trust and uptake.

Weaknesses

It is not entirely clear that the data support the claim that the model passed external validation. While the point estimate for the overall model was just above the AUC cutoff of 0.75, the 95% CIs were (0.67 – 0.84). Furthermore, most of the site specific models has an AUC estimate below 0.75 and this was consistent across various model sets. It is important to note that being below an arbitrary AUC cutoff of 0.75 does not necessarily indicate the model will not perform well in practice.

The operational component is important when interpreting results of CDSS validation studies. In this study the key personnel were general practice nurses / study nurses who enrolled no more than 8 children per day and had no other responsibilities. In actual deployment, one could imagine staff members would likely have other duties in addition to managing kids presenting with acute diarrhea and would need to triage more than 8 per day, depending on the season. This may lead to over-estimation of the CDSS's utility in higher volume / higher acuity settings, perhaps due to entry error or time constraints. The results of the ongoing RCT evaluating this platform will be critical for assessing real-world utility.

From what I could tell, models features are parameterized on a cloud based server and variable weights are then downloaded on to the mobile phone app. I understand the CDSS recommendations were not reported to treating clinicians but were they generated in real time (ie as soon as the staff entered the data)?. How long did it take for results to appear? Would consider describing this a little more clearly. I'm also curious to know how often the model was re-trained on the server (if at all)? Not critical to overall hypothesis but important to think about when trying to develop the infrastructure to deploy this approach in other settings.

Were the study nurses who entered data into the app the same ones who entered the data into the CRF? If so, may be hard to differentiate the App's ease of use and accuracy since there will be correlation in symptom documentation.

The change in the design of the application halfway through the first phase in Bangladesh means that data accrued prior to the change are not relevant to the evaluation of the CDSS' external validity or generalizability. I would consider dropping the '(no date restriction)' analyses from the results.

Unclear what is being referred to with regards to "these independent models" (line 337 – 338). Also, from what I could tell, the prediction model combines 3 separate prediction models in a Bayesian analysis. Why could all features not be included in a single Bayesian model? This is not my area of expertise, but I did not understand the rationale here.

There was significant heterogeneity between the sites in terms of the percentage without a viral etiology and the distribution of organisms. As noted in the limitations section, this may be partially explained by differences in rotavirus seasonality. However, model accuracy seemed paradoxically better in Mali where almost 50% of children did not have an etiology determined. Is this presumably due to better discrimination in cases where bacterial infection is more likely? On a more general note, a deeper (or more explicit) discussion of the factors that could explain the drop in AUC between the internal and external validation studies would be helpful.

Different model sets performed best at the different sites. Might this suggest utility in incorporating a trial period to establish which model has the best performance for a given locale? Might be a better approach than using a single model architecture globally.

Table 4: Should the first row be called 'Patient-Intrinsic Only' to be consistent with Table 3? Also as noted above, would consider dropping the 'no date restriction' results.

Line 490: Probably semantics but the actual model being deployed is a 5 feature logistic regression which was derived from a random forest built on a small but highly curated dataset. This isn't typically what most people think of as machine learning, though technically both fit under the definition.

The time frame for the study is 2 months at each site, ie within one season. It would be helpful to see how well the validation performs across multiple seasons but understand completely the disruptions brought on by the COVID-19 pandemic.

*Reviewer #2:*

A potentially interesting study of 199/302 children <5yo with etiologically defined acute (ie 1-7d) diarrhea in November-December 2019 in Bangladesh and in January-February,2020 in Mali, of whom 22% in Mali and 63% in Bangladesh had only viral etiologies identified, and who could reasonably accurately be predicted to have viral-only diarrhea, using a mobile App. The vast majority of viral etiologies were rotaviral (90/94 in Bangladesh; 24/33 in Mali). However, several concerns and comments include:

1. The ill-explained jargon throughout makes the paper difficult to follow. For example, clearer explanations are especially needed for the "patient intrinsic,"historical patient," recent patient and even viral seasonality and climate models used. Another example is "pre-test odds model" and "viral seasonality model" mentioned in the abstract as their major conclusion without any explanation. So what, then is the take-away message or importance of that (especially given several major limitations noted below)?

2. A glaring gap is the lack of any comment about rotavirus vaccine impact in these populations and for any relevant generalizations made.

3. Were the 73 patients with 'overcalling' of vomiting excluded from all analyses, as this certainly skews any prediction models?

4. It is difficult to believe that there were no norovirus infections in this season in Bangladesh, can this be at least commented upon?

5. The major limitation of only 2 month study periods at each site for highly seasonal pathogen incidences needs comment and explanation.

6. What were the frequencies of antimicrobial use in these study children?

7. The converse hypothesis of potentially treatable bacterial infections (with a single dose of azithromycin for example) or even protozoa such as Giardia, should be examined and mentioned. For example, how many patients were misidentified as 'viral' when they had bacterial or protozoal infections?

[Editors’ note: further revisions were suggested prior to acceptance, as described below.]

Thank you for resubmitting your work entitled "External Validation of a Mobile Clinical Decision Support System for Diarrhea Etiology Prediction in Children: A Multicenter Study in Bangladesh and Mali" for further consideration by *eLife*. Your revised article has been evaluated by David Serwadda (Senior Editor) and a Reviewing Editor.

The manuscript has been improved but there are some remaining issues that need to be addressed, as outlined below:

1) The new Figure 3 could be more clearly presented with a) larger font, b) better labeling (unclear if viral seasonality is seasonality alone or present patient + viral seasonality as in the legend as in the table: I would suggest including all 3), c) inclusion of proportions rather than absolute numbers in the table

2) I am not sure that the App accurately identified viral-only etiology for diarrhea as stated in the abstract. It misclassifies a decent number of cases. To this end, a paragraph is needed in the discussion to impart whether the tool is in fact sufficient for clinical use. I would argue that it is not, based on a relatively low AUC (reasonably high numbers of false negatives and positives). I would suggest that the authors speculate on what additional features could raise the AUC to a level sufficient for use by the bedside.

---

## [Author Response]

Essential revisions:(1) Please address the relative lack of seasonality in the data used for model validation in both cohorts.

Thank you for bringing up this excellent point. As the reviewers point out, given the known seasonal cycles of viral diarrhea infections, ideally, at least one full annual cycle of data would have been collected at each study site to account for seasonal variations in diarrhea epidemiology. However, funding and resource limitations, and the prioritization of including two diverse study sites precluded a longer duration of study enrollment. To discuss these considerations and the potential impact on our findings, we have added the following to the Discussion:

“Additionally, given the known seasonal cycles of viral diarrhea infections, ideally, at least one full annual cycle of data would have been collected at each study site to account for seasonal variations in diarrhea epidemiology. However, funding and resource limitations, and the prioritization of including two diverse study sites precluded a longer duration of study enrollment. Notably, the occurrence of a large seasonal rotavirus peak during the data collection period in Bangladesh may be a reason why the models performed sub-optimally at this site in comparison to Mali. We believe that, had resources been available for a longer duration of study enrollment that accounted for greater seasonal variability, our models would have shown better performance characteristics than in the present analysis. Studies to further validate these models using additional study sites with different seasonal cycles of viral diarrhea over at minimum one full annual cycle are recommended.”

(2) Please discuss how operational components may impact the performance characteristics of the decision making tools.

Thank you for these important suggestions, further details describing how operational components may have impacted the performance of the tool have been added to the following sections.

In Methods, we have clarified that for the purpose of this study all calculations for the primary model deployed on the App were performed on-device (i.e., not on the Cloud server) and models were not re-trained. However, future iterations of the App would be able to use near real-time data input depending on data availability and internet connectivity. We have added:

"All calculations were performed on-device (locally on the smartphone) and allowed App use irrespective of internet connectivity. Calculated probabilities were displayed near instantaneously (within 1 second). The App deployed a single primary model (present patient + viral seasonality as described above) which used data collected at the point-of-care and input by the clinician user, and viral seasonality curves; climate and recent patient data sources were not used in this study because they did not add to the AUC at the GEMS study sites. The models were not re-trained on the server after deployment for this validation study in order to align with standards of clinical practice for AI-enabled clinical decision support currently being set by Boards of Medical Informatics; according to these standards, validated models derived using machine learning should not be subsequently iterated without validation. However, for future iterations of the software that use models requiring near real-time data input (e.g., climate data, recent patient data), the mobile device could access the server if internet connectivity and the necessary data sources were available, and if not available, use data stored locally on the device that is obtained from scheduled backend downloads with the server using the most recent data available.”

As mentioned by Reviewer 1 (“Weaknesses”) an important consideration for the future utility and uptake of this App is the time required to effectively use the app in real-world clinical scenarios. We have added discussion regarding several important operational considerations and recommendations for formal evaluations of usability in future research. We have added to the Discussion:

“Importantly, all data collection for the App was conducted by nurses, the majority of whom had never used a mobile CDSS for clinical use previously. Nurses were able to confidently use the smartphone after a relatively brief training period and enter all necessary data into the tool without difficulty in high-volume clinical settings. While the ease of use or exact time required to enter data into the App was not formally assessed in this validation study, from direct observation nurses were able to enter all required data and obtained calculated outputs within several minutes per patient. Given this time required is similar to that spent with patients during routine clinical care, we believe the App would have high clinical utility in the busy clinical settings for which it was designed. Although the calculations for the primary model deployed on the app were performed on-device, data was transmitted successfully from the App to the Cloud server, from an engineering standpoint this indicates that using real-time data sources is feasible. Additionally, the excellent reliability and agreement between nurses as well as between the App and the paper CRF reference demonstrate that accurate data collection is highly feasible in these clinical settings. Formal evaluation of the usability of the App including an assessment of user experience and metrics such as time required for data entry and for display of calculated outputs using real-time data sources are also needed.”

(3) Please discuss how frequently which the mode missed bacterial or protozoal infections.

Thank you for this suggestion. We have clarified that the aim of the study was not to provide a binary classification decision but rather a continuous predicted risk of viral etiology. However, to illustrate the frequency of misidentified bacterial/protozoal infections, a new figure (Figure 3) has been created showing the number of missed bacterial / protozoal infections using different viral probability cutoffs as suggested by the reviewer.

In Results, we have added:

“While the aim of the study was not to provide a binary classification decision but rather a continuous predicted risk of viral etiology, the numbers of false positives and false negatives at various probability thresholds are shown in Figure 3. Notably, viral seasonality tends to have fewer false negatives while the present patient only model tends to have fewer false positives at various thresholds. Additionally, the sensitivity and specificity of the “present patient” and “present patient” + viral seasonality models were similar (Figure 3).”

(4) Please limit jargon and provide definitions for all terms listed by Reviewer 2. The mechanistic and statistical explanations for α and β in the methods are also insufficient to interpret the results, particularly for the generalist reader. Please provide a lengthier and clearer explanation of the calibration procedure.

Thank you for making these important points, to better clarify terminology and methods, major revisions have been made to the Methods section as below.

The definitions for the terminology used to describe the various models have been clarified and clearly delineated in a new Table 1 entitled, “Model Terminology Definitions and Description”.

We have also greatly expanded our descriptions of model development and the “modular” approach from our prior work published by Brintz et al., (2021) in *eLife*, to orient unfamiliar readers to the background model development needed to contextualize the present study. This section has been significantly revised.

More detailed explanations of the terms “pre-test odds” and “post-test odds” have been added:

“This study team’s prior work described the development of these predictive models that integrates multiple sources of data in a principled statistical framework using a ‘post-test odds formulation.’ This method incorporates observed prior knowledge of a prediction, typically using prior known location-specific prevalence (e.g., historical prevalence of viral diarrhea in Mali), as pre-test odds and then updates the pre-test odds using a single model or series of models based on current data. It also enables electronic real-time updating and flexibility in a “modular” fashion, such that the component models can be flexibly included or excluded according to data availability, an important consideration for LMIC settings in which prior epidemiologic data may be unavailable. The post-test odds formulation combines the likelihood ratios derived from these independent models along with pre-test odds into a single prediction. In order to externally validate the predictions from the post-test odds, we processed the data from this study to match the variables used in previously trained models as closely as possible.”

Lastly, the description of the calibration procedure used to calculate the calibration slope and intercept has expanded in the Methods section, “Calibration was assessed using calibration-in-the-large and calibration slope. The target for calibration slope is 1, where <1 suggests predictions are too extreme and >1 suggests predictions are too moderate. The target for calibration intercept is 0, where negative values suggest overestimation and positive values suggest underestimation.^19^ We estimated the calibration coefficients by regressing predicted values versus the observed proportion of viral cases, calculated using the observed proportion of viral cases within 0.05 plus or minus the predicted probability” (citation Van Calster, B., McLernon, D.J., van Smeden, M. *et al.,* Calibration: the Achilles heel of predictive analytics. *BMC Med*
**17,** 230 (2019). https://doi.org/10.1186/s12916-019-1466-7).

(5) Please discuss the potential impact of the rotavirus vaccine on observed study outcomes.

Thank you for this excellent point. An in-depth discussion of the potential impact of rotavirus vaccine on the model’s current (and future) performance has been added to a new Limitations section. We have clarified that the rotavirus vaccine has not yet been introduced in Bangladesh and therefore do not expect any impact on our results for the Bangladesh study site. However, in Mali, the rotavirus vaccine was introduced in 2014 with most recent available estimated coverage rates from 2019 of 63% among infants 12-23 months. Interestingly, the models performed better at the Mali study site compared to Bangladesh despite fairly high coverage rates of rotavirus vaccine with subsequently much lower rates of rotavirus detected on TAC PCR. These findings may indicate that the clinical features associated with viral diarrhea may be pathogen-independent, and that seasonal variations in viral diarrhea may still be relevant even when overall rates of rotavirus are substantially lower than in prior time periods.

We have added the following to the Limitations section:

“Rotavirus vaccination has not yet been introduced in Bangladesh although a cluster randomized trial was conducted in rural Matlab in 2008-2011, and rotavirus vaccine introduction into the routine immunization program has been planned; thus, we do not expect any effect of the vaccine on the present study results for the Bangladesh study site. However, in Mali, rotavirus vaccination was introduced into the routine immunization program in 2014, as of 2019 official coverage rate was 63% in infants 12-23 months.^40–42^ We recognize that the App may not perform as well in places where the rotavirus vaccine has been introduced given the change in viral diarrhea epidemiology in the years since the data were collected from which the models were trained. However, there remain many countries without access to the rotavirus vaccine, notably in Asia where only a few countries have introduced the vaccine nationally or sub-nationally.^41^ For this reason, the App may be most useful in these settings. Interestingly, the models performed better at the Mali study site compared to Bangladesh despite fairly high coverage rates of rotavirus vaccine with subsequently much lower rates of rotavirus detected on TAC PCR. These findings may indicate that the clinical features associated with viral diarrhea may be pathogen-independent, and that seasonal variations in viral diarrhea may still be relevant even when overall rates of rotavirus are substantially lower than in prior time periods. Lastly, as the rotavirus vaccine is planned to be introduced in an increasing number of countries globally, we also recognize that the model may need to be updated to account for changes in epidemiology as rotavirus vaccine is increasingly introduced and rotavirus becomes a less prominent cause of viral diarrhea.”

Reviewer #1:[…]From what I could tell, models features are parameterized on a cloud based server and variable weights are then downloaded on to the mobile phone app. I understand the CDSS recommendations were not reported to treating clinicians but were they generated in real time (ie as soon as the staff entered the data)?. How long did it take for results to appear? Would consider describing this a little more clearly. I'm also curious to know how often the model was re-trained on the server (if at all)? Not critical to overall hypothesis but important to think about when trying to develop the infrastructure to deploy this approach in other settings.

Thank you for these important points. As mentioned above, we have clarified that for the purpose of this study all calculations for the primary model deployed on the App were performed on-device (i.e., not on the Cloud server) and models were not re-trained. To directly address this comment we have added:

“All calculations were performed on-device (locally on the smartphone) and allowed App use irrespective of internet connectivity. Calculated probabilities were displayed near instantaneously (within 1 second). The App deployed a single primary model (present patient + viral seasonality as described above) which used data collected at the point-of-care and input by the clinician user, and viral seasonality curves; climate and recent patient data sources were not used in this study because they did not add to the AUC at the GEMS study sites. The models were not re-trained on the server after deployment for this validation study in order to align with standards of clinical practice for AI-enabled clinical decision support currently being set by Boards of Medical Informatics; according to these standards, validated models derived using machine learning should not be subsequently iterated without validation. However, for future iterations of the software that use models requiring near real-time data input (e.g., climate data, recent patient data), the mobile device could access the server if internet connectivity and the necessary data sources were available, and if not available, use data stored locally on the device that is obtained from scheduled backend downloads with the server using the most recent data available.”

Were the study nurses who entered data into the app the same ones who entered the data into the CRF? If so, may be hard to differentiate the App's ease of use and accuracy since there will be correlation in symptom documentation.

We have clarified that the study nurses who entered data into the app were the same as those who entered data on the case report forms (CRFs). The purpose of evaluating this correlation was less to examine the usability of the app (which would need to be studied formally using ‘think aloud’ methods or qualitative research evaluating user experience for instance) but rather to ensure from an engineering standpoint that successful data input and transmission from the user (the study nurse) to the phone and to the server (which would be required for future implementation of the App if using real-time data sources) was feasible.

The change in the design of the application halfway through the first phase in Bangladesh means that data accrued prior to the change are not relevant to the evaluation of the CDSS' external validity or generalizability. I would consider dropping the '(no date restriction)' analyses from the results.

We thank the reviewer for this suggestion, however after consideration, we believe that presenting all the results (i.e., with and without date restriction) are important for the reader to understand. As mentioned by the reviewer, the primary analysis includes only the data collected when the vomiting question asked correctly, as this is how the App is intended to be used. However, the results presented demonstrating a substantial drop in the AUC may be valuable to the reader, for comparison and to highlight the potential impact of misunderstanding of the App questions in real-world scenarios. This is also important to underscore the challenges of implementing mHealth apps in diverse contexts, and the need for clarity in how questions are understood by end-users.

We have added the following to Discussion:

“Additionally, given the substantial drop in AUC when the vomiting question was asked incorrectly, a trial period to ensure precise understanding of the App’s questions among the target population would also be prudent. These findings underscore the challenges of implementing mHealth tools in diverse contexts, and the importance of ensuring cross-cultural understanding of questions by clinician and patient end-users in real-world scenarios.”

Unclear what is being referred to with regards to "these independent models" (line 337 – 338). Also, from what I could tell, the prediction model combines 3 separate prediction models in a Bayesian analysis. Why could all features not be included in a single Bayesian model? This is not my area of expertise, but I did not understand the rationale here.

Thank you for bringing up this important point for clarification. We agree the model descriptions and “modular” approach were insufficiently described in our initial submission, and as described above have made major revisions to our Methods section describing the models (see response to Essential Revisions #4). We clarify the reason not all features were included in a single model was to allow for a “modular” design that is customizable based on the availability of data. In many LMICs, data such as viral seasonality curves, historical patient data or weather/climate data may be unavailable. This flexible design thus allows users to customize the most appropriate model to their unique context and target population. The line referenced, “these independent models” has therefore been removed.

There was significant heterogeneity between the sites in terms of the percentage without a viral etiology and the distribution of organisms. As noted in the limitations section, this may be partially explained by differences in rotavirus seasonality. However, model accuracy seemed paradoxically better in Mali where almost 50% of children did not have an etiology determined. Is this presumably due to better discrimination in cases where bacterial infection is more likely? On a more general note, a deeper (or more explicit) discussion of the factors that could explain the drop in AUC between the internal and external validation studies would be helpful.

Thank you for these astute observations, we agree with the reviewer and likewise suspect that a primary reason for the discrepancy in attributable pathogens between the two study sites was due to the very high prevalence of rotavirus (which may be easier to attribute using AFe) during the study enrollment period due to an outbreak in the Bangladesh population at the time. We have added to the Limitations section:

“The higher proportion of non-viral pathogens in Mali may have enabled better discrimination between viral and non-viral pathogens leading to better model performance at the Mali study site.”

We have added additional description to Limitations regarding the drop in AUC between internal and external validation as suggested:

“Changes over time in the epidemiology of diarrhea at the study sites as well as the internal validation being conducted in a very large dataset (>20,000 patients) from the GEMS study compared to this study with 300 patients over a short time period during a surge of rotavirus likely also contributed to the drop in AUC between the internal validation and external validation.”

Different model sets performed best at the different sites. Might this suggest utility in incorporating a trial period to establish which model has the best performance for a given locale? Might be a better approach than using a single model architecture globally.

Thank you for this excellent suggestion. We agree that a trial period would be advisable prior to implementation of the app to determine which model has the best performance for the selected site. This approach also takes advantage of the “modular” component of the models in which different versions of the model can be selected and tailored to obtain the best performance and based on data availability. This has been added to the Discussion:

“A unique aspect of this study was the use of a modular approach to clinical prediction models. In this validation study, the version of the App deployed used a primary model of present patient + viral seasonality data based on the best performing model in internal validation studies. However, the great strength of the modular approach is that it allows for customization according to data availability and local diarrhea epidemiology. In this validation study, we found that the model performance varied depending on the study site. To that end, when possible, a trial period would be prudent prior to implementation of the App to determine which model has the best performance for the selected site.”

Table 4: Should the first row be called 'Patient-Intrinsic Only' to be consistent with Table 3? Also as noted above, would consider dropping the 'no date restriction' results.

The table referenced (now named Table 5) has been revised. Of note, the “Patient-Intrinsic” model has been renamed as “Present Patient” model.

Line 490: Probably semantics but the actual model being deployed is a 5 feature logistic regression which was derived from a random forest built on a small but highly curated dataset. This isn't typically what most people think of as machine learning, though technically both fit under the definition.

Thank you for suggesting this clarification. We have replaced the term “machine learning” with “a Bayesian approach.”

The time frame for the study is 2 months at each site, ie within one season. It would be helpful to see how well the validation performs across multiple seasons but understand completely the disruptions brought on by the COVID-19 pandemic.

As described above, ideally, at least one full annual cycle of data would have been collected at each study site to account for seasonal variations in diarrhea epidemiology. However, funding and resource limitations of this pilot study, and the prioritization of including two diverse study sites precluded a longer duration of study enrollment. We agree and recommend that the models should be validated across multiple seasons in future studies.

Reviewer #2:A potentially interesting study of 199/302 children <5yo with etiologically defined acute (ie 1-7d) diarrhea in November-December 2019 in Bangladesh and in January-February,2020 in Mali, of whom 22% in Mali and 63% in Bangladesh had only viral etiologies identified, and who could reasonably accurately be predicted to have viral-only diarrhea, using a mobile App. The vast majority of viral etiologies were rotaviral (90/94 in Bangladesh; 24/33 in Mali). However, several concerns and comments include:1. The ill-explained jargon throughout makes the paper difficult to follow. For example, clearer explanations are especially needed for the "patient intrinsic,"historical patient," recent patient and even viral seasonality and climate models used. Another example is "pre-test odds model" and "viral seasonality model" mentioned in the abstract as their major conclusion without any explanation. So what, then is the take-away message or importance of that (especially given several major limitations noted below)?

We appreciate these important suggestions. As further detailed above, definitions for model terminology have been explicitly stated in a new table (Table 1) and the Methods section majorly revised (see response to Essential Revisions #4). Additionally, for clarity and to better align this paper’s terminology with the initial model development paper (Brintz et al., 2021) published in *eLife*, we have also changed the term “patient intrinsic” to “present patient” as was used in the initial paper. Lastly, the methods section of the abstract has been revised to clarify the terms referenced in the abstract’s Results section as suggested.

2. A glaring gap is the lack of any comment about rotavirus vaccine impact in these populations and for any relevant generalizations made.

Thank you for this important point, as described above in response to Essential Revisions #5, an in-depth discussion of the potential impact of rotavirus vaccine on the model performance has been added to Limitations.

3. Were the 73 patients with 'overcalling' of vomiting excluded from all analyses, as this certainly skews any prediction models?

Thank you for raising this question, we have clarified that the 73 patients with “overcalling” of vomiting (i.e., when the question was misunderstood to include regurgitation after feeding) were excluded from the primary analysis. However, in secondary analysis presented in Table 5 (no date restriction) the data in which the question was asked incorrectly has been included for comparison and to highlight the potential impact of misunderstanding of the required App input questions in real-world scenarios.

4. It is difficult to believe that there were no norovirus infections in this season in Bangladesh, can this be at least commented upon?

Thank you for this astute observation. We have added our hypotheses on possible reasons why there were no norovirus infections detected in Limitations:

“Notably, there were no norovirus infections detected using these pre-determined thresholds. This may have been due to the particularly high prevalence of other pathogens especially rotavirus in Bangladesh during this time period, the previously documented lower prevalence of norovirus in Bangladesh in November-January compared to other months, as well as the relatively short period of participant enrollment, although using a different AFe cutoff may have detected possible cases of norovirus.”

5. The major limitation of only 2 month study periods at each site for highly seasonal pathogen incidences needs comment and explanation.

Thank you for bringing up this important point. As described above, funding and resource limitations of this pilot study, and the prioritization of including two diverse study sites precluded a longer duration of study enrollment. Further comment and explanations as listed above (see response to Essential Revisions #1) have been added to the Discussion section.

6. What were the frequencies of antimicrobial use in these study children?

Data on reported prior antibiotic use have been added to the Results section:

“Antibiotic use prior to health facility presentation was common with 66% of participants in Bangladesh reporting antibiotic use for the current illness (most commonly azithromycin, ciprofloxacin, and metronidazole alone or in combination) and 13.1% of participants Mali (most commonly cotrimoxazole, amoxicillin, metronidazole).”

7. The converse hypothesis of potentially treatable bacterial infections (with a single dose of azithromycin for example) or even protozoa such as Giardia, should be examined and mentioned. For example, how many patients were misidentified as 'viral' when they had bacterial or protozoal infections?

Thank you for making this important point. As above, we have added a new figure (Figure 3) showing the frequency of misidentified bacterial/protozoal infections using different viral probability cutoffs as suggested by the reviewer.

In Results, we have added:

“While the aim of the study was not to provide a binary classification decision but rather a calculated predicted risk of viral etiology, the numbers of false positives and false negatives at various probability thresholds are shown in Figure 3. Notably, viral seasonality tends to have less false negatives while the present patient only model tends to have less false positives at various thresholds. Additionally, the sensitivity and specificity of the “present patient” and “present patient” + viral seasonality models were similar (Figure 3).”

[Editors’ note: further revisions were suggested prior to acceptance, as described below.]

The manuscript has been improved but there are some remaining issues that need to be addressed, as outlined below:1) The new Figure 3 could be more clearly presented with a) larger font, b) better labeling (unclear if viral seasonality is seasonality alone or present patient + viral seasonality as in the legend as in the table: I would suggest including all 3), c) inclusion of proportions rather than absolute numbers in the table

Figure 3 has been updated and revised as requested with font size and labels adjusted and proportions listed in addition to absolute numbers.

2) I am not sure that the App accurately identified viral-only etiology for diarrhea as stated in the abstract. It misclassifies a decent number of cases. To this end, a paragraph is needed in the discussion to impart whether the tool is in fact sufficient for clinical use. I would argue that it is not, based on a relatively low AUC (reasonably high numbers of false negatives and positives). I would suggest that the authors speculate on what additional features could raise the AUC to a level sufficient for use by the bedside.

Thank you for bringing up this important point, however we argue that despite a modest AUC, there is still great potential clinical utility of our tool which has a similar AUC to that of one of the most widely used clinical prediction models for an infectious disease (Centor score which had an AUC of 0.72 in two external validation studies). Additionally, given that according to WHO guidelines, the majority of diarrhea cases do not warrant antibiotics, even a prediction rule with moderate performance would help to reduce inappropriate antibiotic use in settings where unguided antibiotic use is widespread. We have added a paragraph to the Discussion section (lines 607-617) explaining our justification that the tool will have clinical utility for these reasons.